# Deep Inverse Q-learning with Constraints

**Gabriel Kalweit**[*]
Neurorobotics Lab
University of Freiburg
kalweitg@cs.uni-freiburg.de

**Maria Huegle**[*]
Neurorobotics Lab
University of Freiburg
hueglem@cs.uni-freiburg.de

**Moritz Werling**
BMWGroup
Germany
Moritz.Werling@bmw.de

**Joschka Boedecker**
Neurorobotics Lab and BrainLinks-BrainTools
University of Freiburg
jboedeck@cs.uni-freiburg.de

## Abstract

Popular Maximum Entropy Inverse Reinforcement Learning approaches require the computation of expected state visitation frequencies for the optimal policy under an estimate of the reward function. This usually requires intermediate value estimation in the inner loop of the algorithm, slowing down convergence considerably. In this work, we introduce a novel class of algorithms that only needs to solve the MDP underlying the demonstrated behavior *once* to recover the expert policy. This is possible through a formulation that exploits a probabilistic behavior assumption for the demonstrations within the structure of Q-learning. We propose Inverse Action-value Iteration which is able to fully recover an underlying reward of an external agent in *closed-form* analytically. We further provide an accompanying class of sampling-based variants which do not depend on a model of the environment. We show how to extend this class of algorithms to continuous state-spaces via function approximation and how to estimate a corresponding action-value function, leading to a policy as close as possible to the policy of the external agent, while optionally satisfying a list of predefined hard constraints. We evaluate the resulting algorithms called Inverse Action-value Iteration, Inverse Q-learning and Deep Inverse Q-learning on the Objectworld benchmark, showing a speedup of up to several orders of magnitude compared to (Deep) Max-Entropy algorithms. We further apply Deep Constrained Inverse Q-learning on the task of learning autonomous lane-changes in the open-source simulator SUMO achieving competent driving after training on data corresponding to 30 minutes of demonstrations.

## 1   Introduction

Inverse Reinforcement Learning (IRL) [4] is a popular approach to imitation learning which generally reduces the problem of recovering a demonstrated behavior to the recovery of a reward function that induces the observed behavior, assuming that the demonstrator was (softly) maximizing its long-term return. Previous work solved this problem with different approaches, such as Linear IRL [19, 1] and Large-Margin Q-Learning [21]. A very influential approach which addresses the inherent ambiguity of possible reward functions and induced policies for an observed behavior is Maximum Entropy Inverse Reinforcement Learning (MaxEnt IRL, [26]), a probabilistic formulation of the problem which keeps the distribution over actions as non-committed as possible. This approach has been extended with deep networks as function approximators for the reward function in [25] and

---

[*]Equal Contribution.

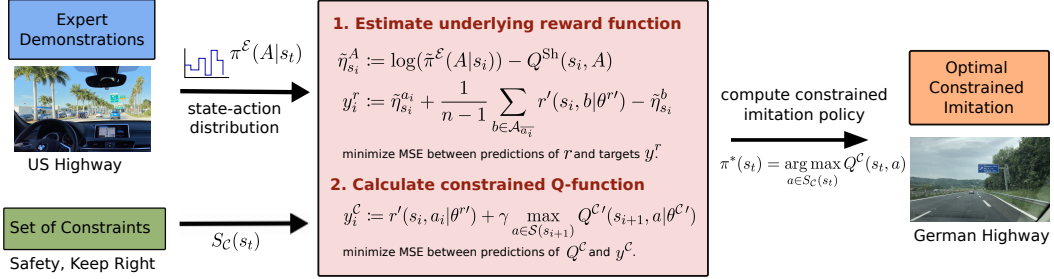

Figure 1: Scheme of Deep Constrained Inverse Q-learning for the constrained transfer of unconstrained driving demonstrations, leading to optimal constrained imitation.

is the basis for several more recent algorithms that lift some of its assumptions (e.g. [8, 10]). One general limitation of MaxEnt IRL based methods, however, is that the considered MDP underlying the demonstrations has to be solved many times inside the inner loop of the algorithm. We also use a probabilistic problem formulation, but assume a policy that only maximizes the entropy over actions locally at each step as an approximation. This leads to a novel class of IRL algorithms based on Inverse Action-value Iteration (IAVI) and allows us to avoid the computationally expensive inner loop (a property shared e.g. with [5]). With this formulation, we are able to calculate a matching reward function for the observed (optimal) behavior *analytically in closed-form*, assuming that the data was collected from an expert following a stochastic policy with an underlying Boltzmann distribution over optimal Q-values, similarly to [18, 20]. Our approach, however, transforms the IRL problem to solving a system of linear equations. Consequently, if there exists an unambiguous reverse topological order and terminal states in the MDP, our algorithm has to solve this system of equations only once for each state, and can achieve speedups of up to several orders of magnitude compared to MaxEnt IRL variants for general infinite horizon problems.

We extend IAVI to a sampling based approach using stochastic approximation, which we call Inverse Q-learning (IQL), using Shifted Q-functions proposed in [12] to make the approach model-free. In contrast to other proposed model-free IRL variants such as Relative Entropy IRL [5] that assumes rewards to be a linear combination of features or Guided Cost Learning [8] that still needs an expensive inner sampling loop to optimize the reward parameters, our proposed algorithm accommodates arbitrary nonlinear reward functions such as neural networks, and needs to solve the MDP underlying the demonstrations only once. A different approach to model-free IRL is Generative Adversarial Imitation Learning (GAIL), recently proposed in [10]. GAIL generates a policy to match the experts behavior without the need to first reconstruct a reward function. In some contexts, however, this can be undesirable, e.g. if additional constraints should be enforced that were not part of the original demonstrations. This is in fact what we cover in a further extension of our algorithms, leading to Constrained Inverse Q-learning (CIQL). Closely related to this is learning by imitation with preferences and constraints as studied in a teacher-learner setting in [23], but the approach internally relies on value estimation with a given transition model and assumes a linear reward formulation. The model-based Inverse KKT approach presented in [7] also studies imitation in a constrained setting, but from the perspective of identifying constraints in the demonstrated behavior. Our final contribution extends all algorithms to the case of continuous state representations by using function approximation, leading to the Deep Inverse Q-Learning (DIQL) and Deep Constrained Inverse Q-learning (DCIQL) algorithms (see Figure 1 for a schematic overview). A compact summary comparing the different properties of the various approaches mentioned above can be found in Table 1. To evaluate our algorithms, we compare the performance for the Objectworld benchmark to MaxEnt IRL based baselines, and present results of an imitation learning task in a simulated automated driving setting where the agent has to learn a constrained lane-change behavior from unconstrained demonstrations. We fix notation and define the IRL problem in Section 2, derive IAVI in Section 3 and its variants in Sections 4-6. Section 7 presents our experimental results and Section 8 concludes.

## 2   Reinforcement Learning and Inverse Reinforcement Learning

We model tasks in the reinforcement learning (RL) framework, where an agent acts in an environment following a policy $\pi$ by applying action $a_t \sim \pi$ from $n$-dimensional action-space $\mathcal{A}$ in state $s_t$

Table 1: Overview of the different IRL approaches.

|  | State-spaces | Model-free | Rewards | Constraints | Inner Loop |
|---|---|---|---|---|---|
| MaxEnt IRL [26] | discrete | × | linear | × | VI |
| Deep MaxEnt IRL [25] | discrete | × | non-linear | × | VI |
| RelEnt IRL [5] | continuous | ✓ | linear | × | × |
| GCL [8] | continuous | ✓ | non-linear | × | PO |
| GAIL [10] | continuous | ✓ | × | × | × |
| AWARE-CMDP [23] | discrete | × | linear | ✓ | VI |
| **IAVI (ours)** | discrete | × | non-linear | × | × |
| **(C)IQL (ours)** | discrete | ✓ | non-linear | (✓) | × |
| **D(C)IQL (ours)** | continuous | ✓ | non-linear | (✓) | × |

from state-space $\mathcal{S}$ in each time step $t$. According to model $\mathcal{M} : \mathcal{S} \times \mathcal{A} \times \mathcal{S} \mapsto [0, 1]$, the agent reaches some state $s_{t+1}$. For every transition, the agent receives a scalar reward $r_t$ from reward function $r : \mathcal{S} \times \mathcal{A} \mapsto \mathbb{R}$ and has to adjust its policy $\pi$ so as to maximize the expected long-term return $R(s_t) = \sum_{i >= t} \gamma^{i-t} r_i$, where $\gamma \in [0, 1]$ is the discount factor. We focus on off-policy Q-learning [24], where an optimal policy can be found on the basis of a given transition set. The Q-function $Q^\pi(s_t, a_t) = \mathbf{E}_{\pi, \mathcal{M}}[R(s_t)|a_t]$ represents the value of an action $a_t$ and following $\pi$ thereafter. From the optimal action-value function $Q^*$, the optimal policy $\pi^*$ can be extracted by choosing the action with highest value in each time step. In the IRL framework, the immediate reward function is unknown and has to be estimated from observed trajectories collected by expert policy $\pi^{\mathcal{E}}$.

## 3 Inverse Action-value Iteration

We start with the derivation of model-based Inverse Action-value Iteration, where we first establish a relationship between Q-values of a state action-pair $(s, a)$ and the Q-values of all other actions in this state. We assume that the trajectories are collected by an agent following a stochastic policy with an underlying Boltzmann distribution according to its unknown optimal value function $Q^*$:

$$\frac{\exp(Q^*(s, a))}{\sum_{A \in \mathcal{A}} \exp(Q^*(s, A))} := \pi^{\mathcal{E}}(a|s), \tag{1}$$

for all actions $a \in \mathcal{A}$. Rearranging gives:

$$\sum_{A \in \mathcal{A}} \exp(Q^*(s, A)) = \frac{\exp(Q^*(s, a))}{\pi^{\mathcal{E}}(a|s)} \text{ and } \exp(Q^*(s, a)) = \pi^{\mathcal{E}}(a|s) \sum_{A \in \mathcal{A}} \exp(Q^*(s, A)). \tag{2}$$

Denoting the set of actions excluding action $a$ as $\mathcal{A}_{\bar{a}}$, we can express the Q-values for an action $a$ in terms of Q-values for any action $b \in \mathcal{A}_{\bar{a}}$ in state $s$ and the probabilities for taking these actions:

$$\exp(Q^*(s, a)) = \pi^{\mathcal{E}}(a|s) \sum_{A \in \mathcal{A}} \exp(Q^*(s, A)) = \frac{\pi^{\mathcal{E}}(a|s)}{\pi^{\mathcal{E}}(b|s)} \exp(Q^*(s, b)). \tag{3}$$

Taking the log[1] then leads to:

$$Q^*(s, a) = Q^*(s, b) + \log(\pi^{\mathcal{E}}(a|s)) - \log(\pi^{\mathcal{E}}(b|s)). \tag{4}$$

Thus, we can relate the optimal value of action $a$ with the optimal values of all other actions in the same state by including the respective log-probabilities. Since this holds for all actions $b \in \mathcal{A}_{\bar{a}}$ (where $|\mathcal{A}_{\bar{a}}| = n - 1$), taking the sum over all actions in $\mathcal{A}_{\bar{a}}$ leads to:

$$(n - 1)Q^*(s, a) = \sum_{b \in \mathcal{A}_{\bar{a}}} Q^*(s, b) + \log(\pi^{\mathcal{E}}(a|s)) - \log(\pi^{\mathcal{E}}(b|s)) \tag{5}$$

$$= (n - 1) \log(\pi^{\mathcal{E}}(a|s)) + \sum_{b \in \mathcal{A}_{\bar{a}}} Q^*(s, b) - \log(\pi^{\mathcal{E}}(b|s)). \tag{6}$$

The optimal action-values for state $s$ and action $a$ are composed of immediate reward $r(s, a)$ and the optimal action-value for the next state as given by the transition model, i.e. $Q^*(s, a) = r(s, a) + \gamma \max_{a'} \mathbf{E}_{s' \sim \mathcal{M}(s,a,s')}[Q^*(s', a')]$. Using this definition in Equation (6) to replace the Q-values and defining the difference between the log-probability and the discounted value of the next state as:

$$\eta_s^a := \log(\pi^{\mathcal{E}}(a|s)) - \gamma \max_{a'} \mathbf{E}_{s' \sim \mathcal{M}(s,a,s')}[Q^*(s', a')], \tag{7}$$

we can then solve for the immediate reward:

$$r(s, a) = \eta_s^a + \frac{1}{n-1} \sum_{b \in \mathcal{A}_{\bar{a}}} r(s, b) - \eta_s^b. \tag{8}$$

Formulating Equation (8) for all actions $a_i \in \mathcal{A}$ results in a system of linear equations $\mathcal{X}_{\mathcal{A}}(s)\mathcal{R}_{\mathcal{A}}(s) = \mathcal{Y}_{\mathcal{A}}(s)$, with reward vector $\mathcal{R}_{\mathcal{A}}(s)$, coefficient matrix $\mathcal{X}_{\mathcal{A}}(s)$ and target vector $\mathcal{Y}_{\mathcal{A}}(s)$:

$$\begin{bmatrix} 1 & -\frac{1}{n-1} & \cdots & -\frac{1}{n-1} \\ -\frac{1}{n-1} & 1 & \cdots & -\frac{1}{n-1} \\ \vdots & \vdots & \ddots & \vdots \\ -\frac{1}{n-1} & -\frac{1}{n-1} & \cdots & 1 \end{bmatrix} \begin{bmatrix} r(s, a_1) \\ r(s, a_2) \\ \vdots \\ r(s, a_n) \end{bmatrix} = \begin{bmatrix} \eta_s^{a_1} - \frac{1}{n-1} \sum_{b \in \mathcal{A}_{\overline{a_1}}} \eta_b^s \\ \eta_s^{a_2} - \frac{1}{n-1} \sum_{b \in \mathcal{A}_{\overline{a_2}}} \eta_b^s \\ \vdots \\ \eta_s^{a_n} - \frac{1}{n-1} \sum_{b \in \mathcal{A}_{\overline{a_n}}} \eta_b^s \end{bmatrix}. \tag{9}$$

**Theorem 1.** *There always exists a solution for the linear system provided by $\mathcal{X}_{\mathcal{A}}(s)$ and $\mathcal{Y}_{\mathcal{A}}(s)$ (proof in the appendix).*

Intuitively, this formulation of the immediate reward encodes the local probability of action $a$ while also ensuring the probability of the maximizing next action under Q-learning. Hence, we note that this formulation of bootstrapping visitation frequencies bears a strong resemblance to the Successor Feature Representation [6, 15]. The Q-function can then be updated via the standard state-action Bellman optimality equation $Q^*(s, a) = r(s, a) + \gamma \max_{a'} \mathbf{E}_{s' \sim \mathcal{M}(s,a,s')}[Q^*(s', a')]$, for all states $s$ and actions $a$. Since $Q^*$ is unknown, however, we cannot estimate $r(s, a)$ directly. We circumvent this necessity by estimating $r(s', a)$ for all terminal states $s'$, i.e. states for which no next state exists. Going through the MDP once in reverse topological order based on its model $\mathcal{M}$, we can compute $Q^*$ for the succeeding states and actions, leading to a reward function for which the induced optimal action-value function yields a Boltzmann distribution matching the true distribution of actions *exactly*. Hence, if the observed transitions are samples from the true optimal Boltzmann distribution, we can recover the *true* reward function of the MDP in *closed-form*. Please note, however, that the expert demonstrations need not necessarily follow a Boltzmann distribution, but can rather follow an *arbitrary* distribution, including multimodal ones and also greedy behavior choice by adding a small conditioning term $\epsilon \ll 1$. Our approach encodes this underlying arbitrary distribution of the expert in the long-term Q-value, such that a Boltzmann distribution over the estimated Q-function is equivalent to the original arbitrary expert distribution.

**Theorem 2.** *Define $Q^*(s, a)$ to satisfy the equation $Q^*(s, a) = Q^*(s, b) + \log(\pi^{\mathcal{E}}(a|s)) - \log(\pi^{\mathcal{E}}(b|s))$ for all actions $a, b \in \mathcal{A}$ and expert policy $\pi^{\mathcal{E}}(\cdot|s)$ of arbitrary non-zero distribution. Then the Boltzmann distribution over $Q^*(s, \cdot)$ is equivalent to $\pi^{\mathcal{E}}(\cdot|s)$.*

*Proof.* The theorem follows from the inverse application of Equations (1)-(3). With $Q^*(s, a) = Q^*(s, b) + \log(\pi^{\mathcal{E}}(a|s)) - \log(\pi^{\mathcal{E}}(b|s))$ for all actions $a, b \in \mathcal{A}$, it follows from Eq. (3) that $\exp(Q^*(s, a)) = (\pi^{\mathcal{E}}(a|s)/\pi^{\mathcal{E}}(b|s)) \exp(Q^*(s, b)) = \pi^{\mathcal{E}}(a|s) \sum_{A \in \mathcal{A}} \exp(Q^*(s, A))$ and thus $\pi^{\mathcal{E}}(a|s) = \exp(Q^*(s, a))/\sum_{A \in \mathcal{A}} \exp(Q^*(s, A))$. $\square$

Put differently, our assumption is a *restriction over the space of Q-functions*, not expert policies. In case of an infinite control problem or if no clear reverse topological order exists, we solve the MDP by iterating multiple times until convergence. Since the Bellman update is a contraction, the reward values become more accurate with each iteration. Convergence guarantees for IAVI and IQL then follow from the convergence guarantees of Value Iteration and Q-learning under the same conditions. If there are actions with zero probability mass under the expert demonstrations, we add a very small conditioning term $\epsilon$ to the probability to avoid numerical instabilities. This leads us back to the case above with guaranteed convergence, but introduces a small deviation with respect to the match of the expert policy, bounded in dependence of $\epsilon$. Empirically, we show convergence in our experiments in Sections 7.1 and 7.2.

## 4 Tabular Inverse Q-learning

To relax the assumption of an existing transition model and action probabilities, we extend the Inverse Action-value Iteration to a sampling-based algorithm. For every transition $(s, a, s')$, we update the reward function based on Equation (8) using stochastic approximation:

$$r(s, a) \leftarrow (1 - \alpha_r)r(s, a) + \alpha_r \left( \eta_s^a + \frac{1}{n-1} \sum_{b \in \mathcal{A}_{\bar{a}}} r(s, b) - \eta_s^b \right), \qquad (10)$$

with learning rate $\alpha_r$. Additionally, we need a state-action visitation counter $\rho(s, a)$ for state-action pairs to calculate their respective log-probabilities: $\tilde{\pi}^{\mathcal{E}}(a|s) := \rho(s, a) / \sum_{A \in \mathcal{A}} \rho(s, A)$. Therefore, we approximate $\eta_s^a$ by $\tilde{\eta}_s^a := \log(\tilde{\pi}^{\mathcal{E}}(a|s)) - \gamma \max_{a'} \mathbf{E}_{s' \sim \mathcal{M}(s,a,s')}[Q^*(s', a')]$. In order to avoid the need of a model $\mathcal{M}$, we evaluate all other actions via Shifted Q-functions as in [12]:

$$Q^{\mathrm{Sh}}(s, a) := \gamma \max_{a'} \mathbf{E}_{s' \sim \mathcal{M}(s,a,s')}[Q^*(s', a')], \qquad (11)$$

i.e. $Q^{\mathrm{Sh}}(s, a)$ skips the immediate reward for taking $a$ in $s$ and only considers the discounted Q-value of the next state $s'$ for the maximizing action $a'$. We then formalize $\tilde{\eta}_s^a$ as:

$$\tilde{\eta}_s^a := \log(\tilde{\pi}^{\mathcal{E}}(a|s)) - \gamma \max_{a'} \mathbf{E}_{s' \sim \mathcal{M}(s,a,s')}[Q^*(s', a')] = \log(\tilde{\pi}^{\mathcal{E}}(a|s)) - Q^{\mathrm{Sh}}(s, a). \qquad (12)$$

Combining this with the count-based approximation $\tilde{\pi}^{\mathcal{E}}(a|s)$ and updating $Q^{\mathrm{Sh}}$, $r$, and $Q$ via stochastic approximation yields the model-free *Tabular Inverse Q-learning* algorithm (cf. Algorithm 1). We show the more general case of an online algorithm, i.e., the possibility to add transitions during training. The algorithm is simple to transfer to the offline case of estimating the visitation probabilities beforehand, see Algorithm 3 in the appendix.

---

**Algorithm 1:** Tabular Inverse Q-learning

1   initialize $r$, $Q$ and $Q^{\mathrm{Sh}}$ and state-action visitation counter $\rho$
2   **for** $episode = 1..E$ **do**
3      get initial state $s_1$
4      **for** $t = 1..T$ **do**
5         observe action $a_t$ and next state $s_{t+1}$, increment counter $\rho(s_t, a_t) = \rho(s_t, a_t) + 1$
6         get probabilities $\tilde{\pi}^{\mathcal{E}}(a|s_t)$ for state $s_t$ and all $a \in \mathcal{A}$ from $\rho$
7         update $Q^{\mathrm{Sh}}$ by $Q^{\mathrm{Sh}}(s_t, a_t) \leftarrow (1 - \alpha_{\mathrm{Sh}})Q^{\mathrm{Sh}}(s_t, a_t) + \alpha_{\mathrm{Sh}}(\gamma \max_a Q(s_{t+1}, a))$
8         calculate for all actions $a \in \mathcal{A}$: $\tilde{\eta}_{s_t}^a = \log(\tilde{\pi}^{\mathcal{E}}(a|s_t)) - Q^{\mathrm{Sh}}(s_t, a)$
9         update $r$ by $r(s_t, a_t) \leftarrow (1 - \alpha_r)r(s_t, a_t) + \alpha_r(\eta_{s_t}^{a_t} + \frac{1}{n-1} \sum_{b \in \mathcal{A}_{\bar{a}_t}} r(s_t, b) - \eta_{s_t}^b)$
10        update $Q$ by $Q(s_t, a_t) \leftarrow (1 - \alpha_Q)Q(s_t, a_t) + \alpha_Q(r(s_t, a_t) + \gamma \max_a Q(s_{t+1}, a))$

---

## 5 Deep Inverse Q-learning

To cope with continuous state-spaces, we now introduce a variant of IQL with function approximation. We estimate reward function $r$ with function approximator $r(\cdot, \cdot | \theta^r)$, parameterized by $\theta^r$. The same holds for $Q$ and $Q^{\mathrm{Sh}}$, represented by $Q(\cdot, \cdot | \theta^Q)$ and $Q^{\mathrm{Sh}}(\cdot, \cdot | \theta^{\mathrm{Sh}})$ with parameters $\theta^Q$ and $\theta^{\mathrm{Sh}}$. To alleviate the problem of moving targets, we further introduce target networks for $r(\cdot, \cdot | \theta^r)$, $Q(\cdot, \cdot | \theta^Q)$ and $Q^{\mathrm{Sh}}(\cdot, \cdot | \theta^{\mathrm{Sh}})$, denoted by $r'(\cdot, \cdot | \theta^{r'})$, $Q'(\cdot, \cdot | \theta^{Q'})$ and $Q^{\mathrm{Sh}'}(\cdot, \cdot | \theta^{\mathrm{Sh}'})$ and parameterized by $\theta^{r'}$, $\theta^{Q'}$ and $\theta^{\mathrm{Sh}'}$, respectively. Each collected transition $(s_t, a_t, s_{t+1})$, either online or in a fixed batch, is stored in replay buffer $\mathcal{D}$. We then sample minibatches $(s_i, a_i, s_{i+1})_{1 \leq i \leq m}$ from $\mathcal{D}$ to update the parameters of our function approximators. First, we calculate the target for the Shifted Q-function:

$$y_i^{\mathrm{Sh}} := \gamma \max_a Q'(s_{i+1}, a | \theta^{Q'}), \qquad (13)$$

and then apply one step of gradient descent on the mean squared error to the respective predictions, i.e. $\mathcal{L}(\theta^{\mathrm{Sh}}) = \frac{1}{m} \sum_i (Q^{\mathrm{Sh}}(s_i, a_i | \theta^{\mathrm{Sh}}) - y_i^{\mathrm{Sh}})^2$. We approximate the state-action visitation by classifier $\rho(\cdot, \cdot | \theta^\rho)$, parameterized by $\theta^\rho$ and with linear output. Applying the softmax on the outputs of

$\rho(\cdot, \cdot | \theta^\rho)$ then maps each state $s$ to a probability distribution over actions $a_j|_{1 \le j \le n}$. Classifier $\rho(\cdot, \cdot | \theta^\rho)$ is trained to minimize the cross entropy between its induced probability distribution and the corresponding targets, i.e.: $\mathcal{L}(\theta^\rho) = \frac{1}{m} \sum_i -\rho(s_i, a_i) + \log \sum_{j \ne i} \exp \rho(s_i, a_j)$. Given the predictions[2] of $\rho(\cdot, \cdot | \theta^\rho)$, we can calculate targets $y_i^r$ for reward estimation $r(\cdot, \cdot | \theta^r)$ by:

$$y_i^r \coloneqq \tilde{\eta}_{s_i}^{a_i} + \frac{1}{n-1} \sum_{b \in \mathcal{A}_{\overline{a_i}}} r'(s_i, b | \theta^{r'}) - \tilde{\eta}_{s_i}^b, \tag{14}$$

and apply gradient descent on the mean squared error $\mathcal{L}(\theta^r) = \frac{1}{m} \sum_i (r(s_i, a_i | \theta^r) - y_i^r)^2$. Lastly, we can perform a gradient step on loss $\mathcal{L}(\theta^Q) = \frac{1}{m} \sum_i (Q(s_i, a_i | \theta^Q) - y_i^Q)^2$, with targets: $y_i^Q = r'(s_i, a_i | \theta^{r'}) + \gamma \max_a Q'(s_{i+1}, a | \theta^{Q'})$, to update the parameters $\theta^Q$ of $Q(\cdot, \cdot | \theta^Q)$. We update target networks by Polyak averaging, i.e. $\theta^{\mathrm{Sh}'} \leftarrow (1-\tau)\theta^{\mathrm{Sh}'} + \tau\theta^{\mathrm{Sh}}$, $\theta^{r'} \leftarrow (1-\tau)\theta^{r'} + \tau\theta^r$ and $\theta^{Q'} \leftarrow (1-\tau)\theta^{Q'} + \tau\theta^Q$. Details of *Deep Inverse Q-learning* can be found in Algorithm 2.

---

**Algorithm 2:** Fixed Batch Deep Inverse Q-learning

---

**input:** replay buffer $\mathcal{D}$

1  initialize networks $r(\cdot, \cdot | \theta^r)$, $Q(\cdot, \cdot | \theta^Q)$ and $Q^{\mathrm{Sh}}(\cdot, \cdot | \theta^{\mathrm{Sh}})$ and classifier $\rho(\cdot, \cdot | \theta^\rho)$
2  initialize target networks $r'(\cdot, \cdot | \theta^{r'})$, $Q'(\cdot, \cdot | \theta^{Q'})$ and $Q^{\mathrm{Sh}'}(\cdot, \cdot | \theta^{\mathrm{Sh}'})$
3  **for** $iteration = 1..I$ **do**
4       sample minibatch $\mathcal{B} = (s_i, a_i, s_{i+1})_{1 \le i \le m}$ from $\mathcal{D}$
5       minimize MSE between predictions of $Q^{\mathrm{Sh}}$ and $y_i^{\mathrm{Sh}} = \gamma \max_a Q'(s_{i+1}, a | \theta^{Q'})$
6       minimize $\mathcal{CE}$ between predictions of $\rho$ and actions $a_i$
7       get probabilities $\tilde{\pi}^{\mathcal{E}}(a|s_i)$ for state $s_i$ and all $a \in \mathcal{A}$ from $\rho(s_i, a | \theta^\rho)$
8       calculate for all actions $a \in \mathcal{A}$: $\tilde{\eta}_{s_i}^a = \log(\tilde{\pi}^{\mathcal{E}}(a|s_i)) - Q^{\mathrm{Sh}'}(s_i, a | \theta^{\mathrm{Sh}'})$
9       minimize MSE between predictions of $r$ and $y_i^r = \tilde{\eta}_{s_i}^{a_i} + \frac{1}{n-1} \sum_{b \in \mathcal{A}_{\overline{a_i}}} r'(s_i, b | \theta^{r'}) - \tilde{\eta}_{s_i}^b$
10      minimize MSE between predictions of $Q$ and $y_i^Q = r'(s_i, a_i | \theta^{r'}) + \gamma \max_a Q'(s_{i+1}, a | \theta^{Q'})$
11      update target networks $r'$, $Q'$ and $Q^{\mathrm{Sh}'}$

---

## 6  Deep Constrained Inverse Q-learning

Following the definition of Constrained Q-learning in [13], we extend IQL to incorporate a set of constraints $\mathcal{C} = \{c_i : \mathcal{S} \times \mathcal{A} \to \mathbb{R} | 1 \le i \le C\}$ shaping the space of safe actions in each state. We define the safe set for constraint $c_i$ as $S_{c_i}(s) = \{a \in \mathcal{A} | c_i(s, a) \le \beta_{c_i}\}$, where $\beta_{c_i}$ is a constraint-specific threshold, and $S_{\mathcal{C}}(s)$ as the intersection of all safe sets. In addition to the Q-function in IQL, we estimate a constrained Q-function $Q^{\mathcal{C}}$ by:

$$Q^{\mathcal{C}}(s, a) \leftarrow (1 - \alpha_{Q^c})Q^{\mathcal{C}}(s, a) + \alpha_{Q^c}\left(r(s, a) + \gamma \max_{a' \in S_{\mathcal{C}}(s')} Q^{\mathcal{C}}(s', a')\right). \tag{15}$$

For policy extraction from $Q^{\mathcal{C}}$ after Q-learning, only the action-values of the constraint-satisfying actions must be considered. As shown in [13], this formulation of the Q-function optimizes constraint satisfaction on the long-term and yields the optimal action-values for the induced constrained MDP. Put differently, including constraints directly in IQL leads to optimal constrained imitation from unconstrained demonstrations. Analogously to *Deep Inverse Q-learning* in Section 5, we can approximate $Q^{\mathcal{C}}$ with function approximator $Q^{\mathcal{C}}(\cdot, \cdot | \theta^{\mathcal{C}})$ and associated target network $Q^{\mathcal{C}'}(\cdot, \cdot | \theta^{\mathcal{C}'})$. We call this algorithm *Deep Constrained Inverse Q-learning*, see pseudocode in the appendix.

## 7  Experiments

We evaluate the performance of IAVI, IQL and DIQL on the common IRL Objectworld benchmark (Figure 2a) and compare to MaxEnt IRL [26] (closest to IAVI of all entropy-based IRL approaches) and Deep MaxEnt IRL [25] (which is able to recover non-linear reward functions). We then show the potential of constrained imitation by DCIQL on a more complex highway scenario in the open-source traffic simulator SUMO [14] (Figure 2b).

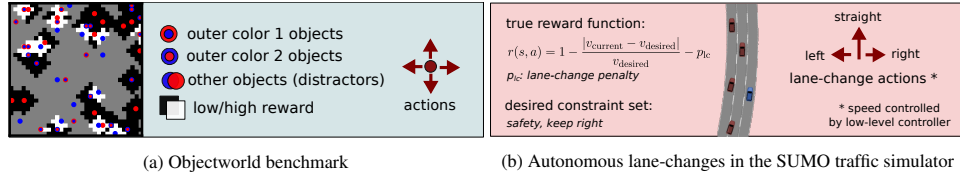

(a) Objectworld benchmark        (b) Autonomous lane-changes in the SUMO traffic simulator

Figure 2: Environments for evaluation of our inverse reinforcement learning algorithms.

## 7.1 Objectworld Benchmark

The Objectworld environment [16] is an $N \times N$ map, where an agent chooses between going *up*, *down*, *left* or *right* or to *stay* in place per time step. Stochastic transitions take the agent in a random direction with 30% chance. Objects are randomly put on the grid with certain inner and outer colors from a set of $C$ colors. In this work, we use the continuous feature representation, which includes $2C$ binary features, indicating the minimum distance to the nearest object with a specific inner and outer color. In the experiments, we use $N = 32$ and $50$ objects and learn on expert trajectories of length $8$. As measure of performance, we use the *expected value difference* (EVD) metric, originally proposed in [16]. It represents how suboptimal the learned policy is under the true reward (see appendix for more details). We compute the state-value under the true reward for the true policy and subtract the state-value under the true reward for the optimal policy w.r.t. the learned reward. Since our derivation assumes the expert to follow a Boltzmann distribution over optimal Q-values, we choose the EVD to be based on the more general stochastic policies. Architectures and hyperparameters are shown in the appendix. We compare the learned state-value function of IAVI, IQL and MaxEnt IRL (vanilla and with a single inner step of VI) trained on a dataset with 1.7M transitions that has an action distribution equivalent to the true underlying Boltzmann distribution for a random Objectworld environment. The results averaged over five training runs are shown in Figure 3. All approaches are trained until convergence (difference of learned reward $< 10^{-4}$ between iterations). IAVI matches the ground truth distribution almost exactly with an EVD of 0.09 (due to the infinite control problem there are slight deviations at this point of convergence), while IQL shows a low mean EVD of 1.47 and the MaxEnt IRL methods 11.58 and 4.33, respectively. Additionally, IAVI and IQL have *tremendously* lower runtimes than MaxEnt IRL, with $1.77 \, \text{min}$ and $21.06 \, \text{min}$ compared to $8.08 \, \text{h}$. This illustrates the dramatic effect of IAVI not needing an inner loop, in contrast to MaxEnt IRL, which has to compute expected state visitation frequencies of the optimal policy repeatedly. We further compare with a variant of MaxEnt IRL with only a single inner step of value-iteration (motivated by the approximation in [8]). Though this variant speeds up the time per iteration considerably, it has a runtime of $12.2 \, \text{h}$ due to a much higher amount of required iterations until convergence. A performance comparison for different numbers of expert demonstrations is shown in Figure 4. While MaxEnt IRL shows the worst performance of all approaches, Deep MaxEnt IRL generalizes very well for 8 to 256 trajectories, but shows high variance and a significant increase of the EVD for 512 trajectories. Because IAVI and IQL are tabular and converging to match the action distribution of the expert exactly, the algorithms need more samples than Deep MaxEnt IRL to achieve an EVD close to 0.0 as the action distribution is

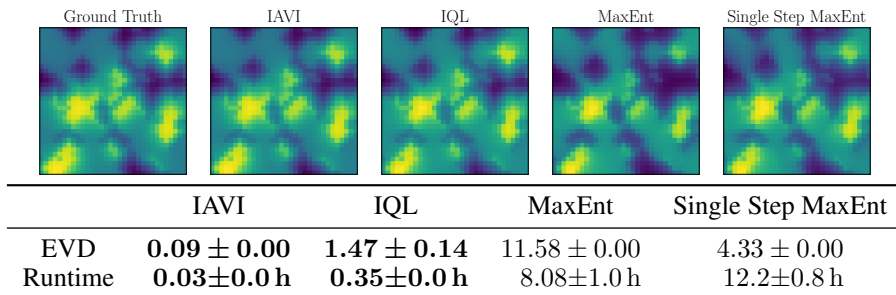

| | IAVI | IQL | MaxEnt | Single Step MaxEnt |
|---|---|---|---|---|
| EVD | $\mathbf{0.09 \pm 0.00}$ | $\mathbf{1.47 \pm 0.14}$ | $11.58 \pm 0.00$ | $4.33 \pm 0.00$ |
| Runtime | $\mathbf{0.03 \pm 0.0 \, h}$ | $\mathbf{0.35 \pm 0.0 \, h}$ | $8.08 \pm 1.0 \, h$ | $12.2 \pm 0.8 \, h$ |

Figure 3: Results for the Objectworld environment, given a data set with an action distribution equivalent to the true optimal Boltzmann distribution. Visualization of the true and learned state-value functions (top). Resulting expected value difference and time needed until convergence, mean and standard deviation over 5 training runs on a $3.00 \, \text{GHz}$ CPU (bottom).

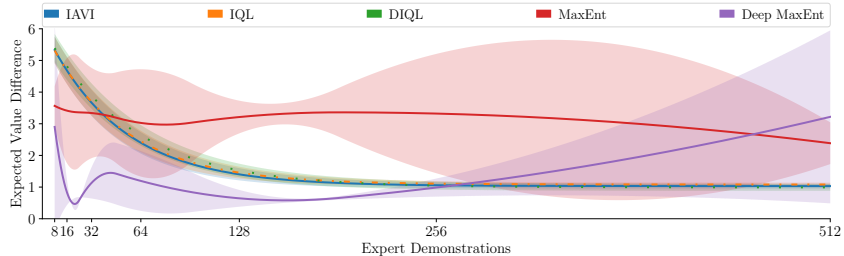

Figure 4: Mean EVD and SD over 5 runs for different numbers of demonstrations in Objectworld.

converging to the true underlying Boltzmann distribution of the optimal policy. Our algorithms start to outperform both MaxEnt methods for more than 256 trajectories and show stable results with low variance. In the Objectworld experiments, we evaluate DIQL using the true action distribution and employ function approximation only to estimate Q- and reward values for fair comparison. Improved generalization can be achieved via approximation of the action distribution, as shown in Section 7.2. We provide results for a greedy expert policy in Objectworld in Figure 5. IAVI outperforms MaxEnt IRL also in this setting by multiple orders of magnitude w.r.t. runtime. IAVI and IQL yield a smaller EVD after less training time. We also address the case of greedy demonstrations in Section 7.2.

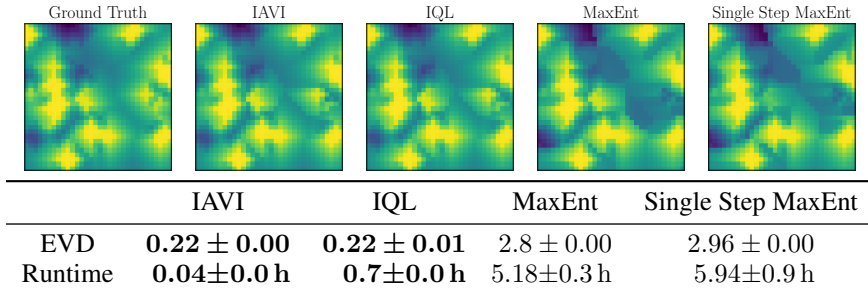

| | IAVI | IQL | MaxEnt | Single Step MaxEnt |
|---|---|---|---|---|
| EVD | **0.22 ± 0.00** | **0.22 ± 0.01** | 2.8 ± 0.00 | 2.96 ± 0.00 |
| Runtime | **0.04±0.0 h** | **0.7±0.0 h** | 5.18±0.3 h | 5.94±0.9 h |

Figure 5: Results for a greedy expert policy in Objectworld, analogously to Figure 3.

## 7.2 High-level Decision Making for Autonomous Driving

We apply Deep Constrained Inverse Q-Learning (DCIQL) to learn autonomous lane-changes on highways from demonstrations. DCIQL allows for long-term optimal constrained imitation while always satisfying a given set of constraints, such as traffic rules. In this setup, we transfer driving styles collected in SUMO from an agent trained without constraints to a different scenario where all vehicles, including the agent, ought to keep right. This corresponds to the task of transferring driving styles from US to German highways by including a keep-right constraint referring to a traffic rule in Germany where drivers ought to drive right when there is a gap of at least $20\,\mathrm{s}$ under the current velocity. We train on highway scenarios with a $1000\,\mathrm{m}$ three-lanes highway and random numbers of vehicles and driver types. For evaluation, we sample 20 scenarios *a priori* for each $n \in (30, 35, \ldots, 90)$ vehicles on track to account for the inherent stochasticity of the task. We use simulator setting, state representation, reward function and action space as proposed in [11]. The state-space consists of the relative distances, velocities and relative lane indices of all surrounding vehicles. The action space consists of a discrete set of actions in lateral direction: *keep lane*, *left lane-change* and *right lane-change*. Acceleration and collision avoidance are controlled by low-level controllers. As expert, we use a fully-trained DeepSet-Q agent as described in [11]. The reward function of the agent encourages driving smoothly and as close as possible to a desired velocity, as depicted in Figure 2b. For all agents and networks, we use the architecture proposed in [11], which can deal with a variable number of surrounding vehicles. Details of the architecture can be found in the appendix. We trained DCIQL on $5 \cdot 10^4$ samples of the expert for $10^5$ iterations. The mean velocities and total number of constraint violations over all scenarios are shown in Figure 6a for the Expert-DQN agent and DCIQL for five training runs. DCIQL (green) shows only a minor loss in performance compared to the expert (red) while satisfying the Keep Right rule, leading to a total

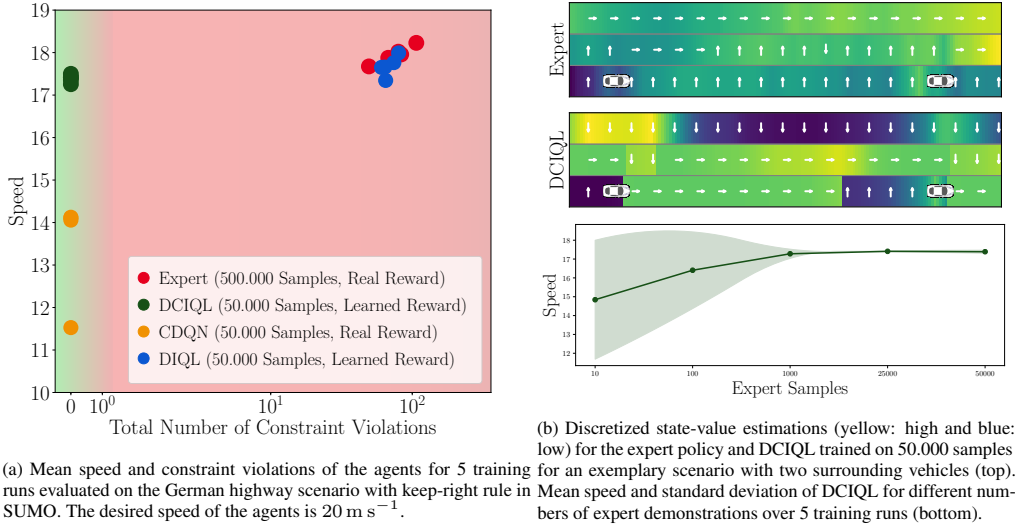

(a) Mean speed and constraint violations of the agents for 5 training runs evaluated on the German highway scenario with keep-right rule in SUMO. The desired speed of the agents is $20\,\mathrm{m\,s^{-1}}$.

(b) Discretized state-value estimations (yellow: high and blue: low) for the expert policy and DCIQL trained on 50.000 samples for an exemplary scenario with two surrounding vehicles (top). Mean speed and standard deviation of DCIQL for different numbers of expert demonstrations over 5 training runs (bottom).

Figure 6: Results for DCIQL in the autonomous driving task.

number of 0 constraint violations for all scenarios. The DIQL agent is imitating the expert agent well and achieves near-equivalent performance whilst also trained for only $10^5$ iterations on $5 \cdot 10^4$ samples. We further trained a CDQN agent as described in [13] in the same manner as the DCIQL agent, but on the true underlying reward function of the MDP. The CDQN agent shows a much worse performance for the same number of iterations. This emphasizes the potential of our learned reward function, offering much faster convergence without the necessity of hand-crafted reward engineering. Compared to our learned reward, the agent trained on the true reward function has a higher demand for training samples and requires more iterations to achieve a well-performing policy. Figure 6b (top) shows the state-value estimations for all possible positions of the agent in a scene of $160\,\mathrm{m}$ with two surrounding vehicles. While the Expert-DQN agent was not trained to include the Keep Right constraint, the DCIQL agent is satisfying the Keep Right and Safety constraints while still imitating to overtake the other vehicles in an anticipatory manner. Figure 6b (bottom) depicts the performance of DCIQL for different numbers of expert demonstrations. DCIQL shows excellent and robust performance with only 1000 provided samples, which corresponds to approximately $1/2\,\mathrm{h}$ of driving demonstrations in simulation. Since the reward learned in DCIQL incorporates long-term information by design, the algorithm is much more data-efficient and thus a very promising approach for the application on real systems.

## 8 Conclusion

We introduced the novel Inverse Action-value Iteration algorithm and an accompanying class of sampling-based variants that provide the first combination of IRL and Q-learning. Our approach needs to solve the MDP underlying the demonstrated behavior only *once* leading to a speedup of up to several orders of magnitude compared to the popular Maximum Entropy IRL algorithm and some of its variants. In addition, it can accommodate arbitrary non-linear reward functions. Extensions of DIQL to continuous action spaces and Soft Q-learning, as well as addressing significant bias in the expert policy are promising directions for future work. Interestingly, our results show that the reward representation we learn through IRL with our approach can lead to significant speedups compared to standard RL training with the true immediate reward function, which we hypothesize to result from the bootstrapping formulation of state-action visitations in our IRL formulation, suggesting a strong link to successor features [6, 15]. We presented model-free variants and showed how to extend the algorithms to continuous state-spaces with function approximation. Our deep constrained IRL algorithm DCIQL is able to guarantee satisfaction of constraints on the long-term for optimal constrained imitation, even if the original demonstrations violate these constraints. For the application of learning autonomous lane-changes on highways, we showed that our approach can learn competent driving strategies satisfying desired constraints at all times from data that corresponds to 30 minutes of driving demonstrations.

## Broader Impact

Our work contributes an advancement in Inverse Reinforcement Learning (IRL) methods that can be used for imitation learning. Importantly, they enable non-expert users to program robots and other technical devices merely by demonstrating a desired behavior. On the one hand, this is a crucial requirement for realising visions such as Industry 4.0, where people increasingly work alongside flexible and lightweight robots and both have to constantly adapt to changing task requirements. This is expected to boost productivity, lower production costs, and contribute to bringing back local jobs that were lost due to globalization strategies. Since the IRL approach we present can incorporate and enforce constraints on behavior even if the demonstrations violate them, it has interesting applications in safety critical applications, such as creating safe vehicle behaviors for automated driving. On the other hand, the improvements we present can potentially accelerate existing trends for automation, requiring less and less human workers if they can be replaced by flexible and easily programmable robots. Estimates for the percentage of jobs at risk for automation range between 14% (OECD report, [17]) and 47% [9] of available jobs (also depending on the country in question). Thus, our work could potentially add to the societal challenge to find solutions that mitigate these consequences (such as e.g. re-training, continuing education, universal basic income, etc.) and make sure that affected individuals remain active, contributing, and self-determined members of society. While it has been argued that IRL methods are essential for value-alignment of artificial intelligence agents [22, 2], the standard framework might not cover all aspects necessary [3]. Our Deep Constrained Inverse Q-learning approach, however, improves on this situation by providing a means to enforce additional behavioral rules and constraints to guarantee behavior imitation consistent within moral and ethical frames of society.

## Disclosure of Funding

This work has been supported by BMWGroup, Germany, and by BrainLinks-BrainTools Cluster of Excellence funded by the German Research Foundation (DFG, grant number EXC 1086). There are no other competing interests.

## Footnotes

[1]For numerical stability, a small $\epsilon \ll 1$ can be added for probabilities equal to 0.

[2]For numerical stability, we clip log-probabilities and update actions if $\tilde{\pi}^{\mathcal{E}}(a|s) > \epsilon$, where $\epsilon \ll 1$.

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
