[Supplementary Material]

# Deep Inverse Q-learning with Constraints
# Appendix

**Gabriel Kalweit**[*]
Neurorobotics Lab
University of Freiburg
kalweitg@cs.uni-freiburg.de

**Maria Huegle**[*]
Neurorobotics Lab
University of Freiburg
hueglem@cs.uni-freiburg.de

**Moritz Werling**
BMWGroup
Germany
Moritz.Werling@bmw.de

**Joschka Boedecker**
Neurorobotics Lab and BrainLinks-BrainTools
University of Freiburg
jboedeck@cs.uni-freiburg.de

## A  Objectworld: Additional Results

Visualizations of the real and learned state-values of IAVI, IQL and DIQL can be found in Figure 7.

Figure 7: Visualization of state-values for different numbers of trajectories in Objectworld.

The approaches match the action distribution of the observed trajectories (denoted as *Ground Truth*) without any visual difference. With increasing number of trajectories, the distributions are converging to the distribution of the optimal policy of the expert.

---

[*]Equal Contribution.

---

**Algorithm 3:** Fixed Batch Tabular Inverse Q-learning

---

**input:** replay buffer $\mathcal{D}$

1  initialize $r$, $Q$ and $Q^{\text{Sh}}$ and state-action visitation counter $\rho$

2  **for** $(s_i, a_i)$ *in* $\mathcal{D}$ **do**

3       increment counter $\rho(s_i, a_i) = \rho(s_i, a_i) + 1$

4  **for** $iteration = 1..I$ **do**

5       sample $(s_i, a_i, s_{i+1})$ from $\mathcal{D}$

6       get probabilities $\tilde{\pi}^{\mathcal{E}}(a|s_i)$ for state $s_i$ and all $a \in \mathcal{A}$ from $\rho$

7       update $Q^{\text{Sh}}$ by $Q^{\text{Sh}}(s_i, a_i) \leftarrow (1 - \alpha_{\text{Sh}})Q^{\text{Sh}}(s_i, a_i) + \alpha_{\text{Sh}}\left(\gamma \max_a Q(s_{i+1}, a)\right)$

8       calculate for all actions $a \in \mathcal{A}$: $\tilde{\eta}^a_{s_i} = \log(\tilde{\pi}^{\mathcal{E}}(a|s_i)) - Q^{\text{Sh}}(s_i, a)$

9       update $r$ by $r(s_i, a_i) \leftarrow (1 - \alpha_r)r(s_i, a_i) + \alpha_r(\eta^{a_i}_{s_i} + \frac{1}{n-1}\sum_{b \in \mathcal{A}_{\overline{a_i}}} r(s_i, b) - \eta^b_{s_i})$

10      update $Q$ by $Q(s_i, a_i) \leftarrow (1 - \alpha_Q)Q(s_i, a_i) + \alpha_Q(r(s_i, a_i) + \gamma \max_a Q(s_{i+1}, a))$

---

A comparison between IQL with online and offline estimated action probabilities is given in Table 2 and a detailed description of Fixed Batch Tabular Inverse Q-learning can be found in Algorithm 3.

Table 2: Comparison between online and offline estimation of state-action visitations for the Objectworld environment, given a data set with an action distribution equivalent to the true optimal Boltzmann distribution. Resulting expected value difference and time needed until convergence, mean and standard deviation over 5 training runs on a 3.00 GHz CPU.

|  | IQL (online $\tilde{\pi}^{\mathcal{E}}$) | IQL (offline $\tilde{\pi}^{\mathcal{E}}$) |
|---|---|---|
| EVD | $1.47 \pm 0.14$ | $1.42 \pm 0.11$ |
| Runtime | $0.35 \pm 0.0\,\text{h}$ | $0.38 \pm 0.0\,\text{h}$ |

## B   Constrained Inverse Q-learning

The pseudocode of the tabular variant of Constrained Inverse Q-learning can be found in Algorithm 4. See [4] for further details of Constrained Q-learning.

---

**Algorithm 4:** Tabular Model-free Constrained Inverse Q-learning

---

1  initialize $r$, $Q$ and $Q^{\text{Sh}}$

2  initialize state-action visitation counter $\rho$

3  initialize $Q^{\mathcal{C}}$ **for** $episode = 1..E$ **do**

4       get initial state $s_1$

5       **for** $t = 1..T$ **do**

6           observe action $a_t$ and next state $s_{t+1}$, increment counter $\rho(s_t, a_t) = \rho(s_t, a_t) + 1$

7           get log-probabilities $\tilde{\pi}^{\mathcal{E}}(a|s_t)$ for state $s_t$ and all $a \in \mathcal{A}$ from $\rho$

8           update $Q^{\text{Sh}}$ by $Q^{\text{Sh}}(s_t, a_t) \leftarrow (1 - \alpha_{\text{Sh}})Q^{\text{Sh}}(s_t, a_t) + \alpha_{\text{Sh}}\left(\gamma \max_a Q(s_{t+1}, a)\right)$

9           calculate for all actions $a \in \mathcal{A}$: $\tilde{\eta}^a_{s_t} = \log(\tilde{\pi}^{\mathcal{E}}(a|s_t)) - Q^{\text{Sh}}(s_t, a)$

10          update $r$ by $r(s_t, a_t) \leftarrow (1 - \alpha_r)r(s_t, a_t) + \alpha_r(\eta^{a_t}_{s_t} + \frac{1}{n-1}\sum_{b \in \mathcal{A}_{\overline{a_t}}} r(s_t, b) - \eta^b_{s_t})$

11          update $Q$ by $Q(s_t, a_t) \leftarrow (1 - \alpha_Q)Q(s_t, a_t) + \alpha_Q(r(s_t, a_t) + \gamma \max_a Q(s_{t+1}, a))$

12          update $Q^{\mathcal{C}}$ by
$$Q^{\mathcal{C}}(s_t, a_t) \leftarrow (1 - \alpha_{\mathcal{C}})Q^{\mathcal{C}}(s_t, a_t) + \alpha_{\mathcal{C}}(r(s_t, a_t) + \gamma \max_{a \in \mathcal{S}(s_{t+1})} Q^{\mathcal{C}}(s_{t+1}, a))$$

---

The pseudocode of Deep Constrained Inverse Q-learning can be found in Algorithm 5. In contrast to classical Deep Inverse Q-learning, the maximization is performed over the safe action set, analogously to [4].

**Algorithm 5:** Fixed Batch Deep Constrained Inverse Q-learning

---

**input:** replay buffer $\mathcal{D}$

1    initialize networks $r(\cdot,\cdot|\theta^r)$, $Q(\cdot,\cdot|\theta^Q)$ and $Q^{\text{Sh}}(\cdot,\cdot|\theta^{\text{Sh}})$ and classifier $\rho(\cdot,\cdot|\theta^\rho)$

2    initialize target networks $r'(\cdot,\cdot|\theta^{r'})$, $Q'(\cdot,\cdot|\theta^{Q'})$ and $Q^{\text{Sh}\prime}(\cdot,\cdot|\theta^{\text{Sh}\prime})$

3    initialize $Q^{\mathcal{C}}(\cdot,\cdot|\theta^{\mathcal{C}})$ and $Q^{\mathcal{C}\prime}(\cdot,\cdot|\theta^{\mathcal{C}\prime})$

4    **for** $iteration = 1..I$ **do**

5       sample minibatch $\mathcal{B} = (s_i, a_i, s_{i+1})_{1 \le i \le m}$ from $\mathcal{D}$

6       minimize MSE between predictions of $Q^{\text{Sh}}$ and $y_i^{\text{Sh}} = \gamma \max_a Q'(s_{i+1}, a|\theta^{Q'})$

7       minimize $\mathcal{CE}$ between predictions of $\rho$ and actions $a_i$

8       get log-probabilities $\tilde{\pi}^{\mathcal{E}}(a|s_i)$ for state $s_i$ and all $a \in \mathcal{A}$ from $\rho(s_i, a|\theta^\rho)$

9       calculate for all actions $a \in \mathcal{A}$: $\tilde{\eta}_{s_i}^a = \log(\tilde{\pi}^{\mathcal{E}}(a|s_i)) - Q^{\text{Sh}\prime}(s_i, a|\theta^{\text{Sh}\prime})$

10      minimize MSE between predictions of $r$ and $y_i^r = \tilde{\eta}_{s_i}^{a_i} + \frac{1}{n-1}\sum_{b \in \mathcal{A}_{\overline{a_i}}} r'(s_i, b|\theta^{r'}) - \tilde{\eta}_{s_i}^b$

11      minimize MSE between predictions of $Q$ and $y_i^Q = r'(s_i, a_i|\theta^{r'}) + \gamma \max_a Q'(s_{i+1}, a|\theta^{Q'})$

12      minimize MSE between $Q^{\mathcal{C}}$ and $y_i^{\mathcal{C}} = r'(s_i, a_i|\theta^{r'}) + \gamma \max_{a \in \mathcal{S}(s_{i+1})} Q^{\mathcal{C}\prime}(s_{i+1}, a|\theta^{\mathcal{C}\prime})$

13      update target networks $r'$, $Q'$, $Q^{\text{Sh}\prime}$ and $Q^{\mathcal{C}\prime}$

---

## C    Objectworld: Reward Function, Architectures and Hyperparameter Optimization

Following the definition of the reward function in [6], states within distance 3 to outer color 1 and distance 2 to outer color 2 yield a positive reward, states within distance 3 to outer color 1 a negative reward and all other states 0 reward, including all other colors which act as distractors.

We optimized the learning rates of the different approaches using grid search over five training runs for random Objectworld environments. For the MaxEnt IRL approaches, we tested learning rates in the interval $(0.1, 0.001)$. We achieved best results with a learning rate of $0.01$. For Deep MaxEnt, we used a 3-layer neural network with 3 hidden dimensions for the approximation of the reward and a learning rate of $0.001$. The learning rates of the table-updates for the reward function, Q-function and the Shifted Q-function for IQL were optimized in a range of $(0.1, 10^{-4})$. The best performing learning rates were $10^{-3}$ for all functions. An extensive evaluation of the influence of the learning rates on the convergence behavior for Online IQL is shown in Figure 8.

Figure 8: Online IQL with different learning rates for an Objectworld environment. In parentheses are the learning rates for $r$, $Q$ and $Q^{\text{Sh}}$, respectively. The upper row shows the KL-divergence between the true and the sample action distribution. The lower row shows the EVD.

For DIQL, the parameters were optimized in the range of $(0.01, 10^{-5})$ with the optimizer Adam [5] and used Rectified Linear Units activations. As final learning rates, we chose $10^{-4}$ for all networks. Target networks are updated with a step-size of $\tau = 10^{-4}$.

## D  SUMO: Architectures and Hyperparameter Optimization

For the function approximators of the reward function, Q-function and Shifted-Q function, we use the hyperparameter-optimized architecture proposed in [3] with two fully-connected layers with $[20, 80]$ hidden dimensions to compute latent-representations of the vehicles with the encoder module $\phi$ and two fully-connected layers for the module $\rho$ with 80 and 20 hidden neurons and two layers with 100 neurons in the Q-network module. We train networks with learning rates of $10^{-4}$. Target networks are updated with a step-size of $\tau = 10^{-4}$. Additionally, we use a variant of Double-Q-learning proposed in [7], which is based on two Q-networks and uses the minimum of the predictions for the target calculation, similar as in [2].

## E  Proof of Theorem 1

**Theorem 1.** *There always exists a solution for the linear system provided by $\mathcal{X}_{\mathcal{A}}(s)$ and $\mathcal{Y}_{\mathcal{A}}(s)$.*

*Proof.* By finding the row echelon form, it can be shown that the rank of the coefficient matrix $\mathcal{X}_{\mathcal{A}}$ is $n-1$. In order to proof that there is at least one solution for the system of linear equations, it has to be shown that the rank of the augmented matrix remains the same. The rank of a matrix cannot decrease by adding a column. Hence, it can only increase. It can easily be shown, that a linear combination of all entries of the augmentation vector $\mathcal{Y}_{\mathcal{A}}$ adds to 0, thus the rank does not increase. Following the Rouché–Capelli theorem [1], there always exists at least one solution, which can then be found with a solver such as least-squares. $\square$