[Reviews · NeurIPS 2020]

Review 1

Summary and Contributions: UPDATE: I thank the authors for their feedback. It seems to me that the authors tried to resolve most of the mentioned issues. Among them, they explain the possible confusion about the boltzmann distribution, the most common criticism of all other reviewers. Ultimately, the decision will depend on other reviewers' stance. The paper presents a new approach to Inverse Reinforcement Learning (IRL) in four algorithms - tabular and model based IAVI, tabular model-free IQL, deep model-free DIQL and finally DCIQL with constraints. The main benefit of the presented algorithms is their order-of-magnitude faster convergence to better results. The authors present definitions of the algorithms and benchmark them in two domains, ObjectWorld and autonomous driving SUMO, making comparison with a related algorithm (Deep) MaxEnt. This is a very clear paper with strong results and high significance, hence I strongly advocate its acceptance.

Strengths: The paper is extremely well written, with elaborated comparison to related work. It reads with ease, mainly due to the ordering of the presented concepts. First presented algorithm results in a system of linear equations that are solvable in closed-form for some domains or numerically (e.g., for infinite MDPs with no terminating states). The subsequent algorithms gradually build on top of each other, making their definition very clear. The algorithms avoid an internal loop of prior algorithm MaxEnt, and as such they are order-of-magnitude more efficient. The first experiment provide evidence for the speed-up, and also stable convergence to a near-precise solution (even in the case of deep DIQL). The second experiment showcases the possibility of incorporating contraints with unconstrained expert demonstrations. Mainly due to the speed-up and convergence properties, the presented algorithms are of high interest to the community. I also appreciated the included video briefly summarizing the paper.

Weaknesses: I found none.

Correctness: Seems correct.

Clarity: Very clear.

Relation to Prior Work: Yes, thoroughly.

Reproducibility: Yes

Additional Feedback: The supplementary material does not contain the source code. For reproducibility, I would appreciate publishing the source code later.


Review 2

Summary and Contributions: [UPDATE] I thank the authors for their response. I agree that the empirical results (for the settings considered in the main paper and the additional results provided in the rebuttal) are convincing/impressive. But in the theoretical/algorithmic front, I'm still not convinced. Especially: [1] lines 4-16 in the rebuttal: I still think that Theorem 1 imposes strong restriction of the class of MDPs (even if it relaxes the restriction on the expert policy distribution): not necessarily all the MDPs should satisfy such condition over long-term Q-value. Consider an example: action space that contains only two actions A = {a,b}, state s, and a greed/deterministic expert policy s.t. \pi(s) = a; then the condition in Theorem 1 is Q^*(s,a) = Q^*(s,b) + \infty ; I don't know how to interpret this condition and how would it affect the following analysis in Section 3. I believe that this needs more discussion to fleshout broader applicability of their claims. [2] lines 26-35 in the rebuttal: this explanation is only partly clear to me. It is better to have explicit formal statements/theorems on covergence guarantee for sections 3 and 4 (as I believe that these two sections are main contributions of this paper; sections 5 and 6 are valued added extensions based on the standard tricks from the literature). Based on the above two points, I expect that it woud require substantial amount of work to revise and would warrant another round of discussion/reviews. ============================= This paper aims to address a fundamental computational problem in standard inverse reinforcement learning algorithms (e.g., MaxEnt): exact computation of the optimal policy w.r.t. current reward parameter at every iteration. To this end, this paper proposes a new algorithm, named inverse-action value iteration, that has to solve a system of equations only once for each state. But their proposal depends on two assumptions: (a) expert policy follows a certain probabilistic form, and (b) MDP should have terminal states and an ordering property. They have provided a sampling variant of this algorithm: inverse q-learning. Then, they extended this algorithm to high dimensional setting using the standard from DQN (e.g., replay buffer, target network, etc.). Finally, they have also a constrained MDP setting as well.

Strengths: The paper proposes a novel algorithm for the IRL problem with the potential to reduce the number of exact policy evaluation steps. The paper is fairly well written, and illustrative experimental results are provided. Reproducibility: Hyperparameter details are clearly documented in Appendix C and D.

Weaknesses: I consider the following as the limitations of/concerns about this work: 1/ What would happen to your strategy if the expert demonstrations are not from Boltzmann policy. Say that the expert policy is soft-optimal policy as in MaxEnt, then the only change in Eq (1), in the place of optimal Q^*, you would use soft-optimal Q^soft (this satisfies soft Bellman equation). How would this affect your strategy in section 3? 2/ At the end of section 3, you state that “In case of an infinite control problem or if no clear reverse topological order exists, we solve the MDP by iterating multiple times until convergence.” Is this procedure guaranteed to converge, and reach policy matching? 3/ Does the tabular inverse Q-learning algorithm guaranteed to converge? What theoretical guarantees you can provide for this algorithm? 4/ Regarding Algorithm 1, line 5: if you compute the state-action visitation frequency (by traversing through the expert data) beforehand, wouldn’t it improve the accuracy of the expert policy estimate in line 6? 5/ In the experiments (section 7.1) are the expert demonstrations generated from: (a) optimal policy w.r.t. true reward or (b) Boltzmann policy as given in Eq (1). If (b) is the case, the comparison of MaxEnt with IQL (your algorithm) may not be fair. Because MaxEnt IRL implicitly assumes that the expert policy is optimal w.r.t. unknown true reward (and you are also evaluating the final performance w.r.t. this reward). Thus to satisfy the assumption required for IQL, you’re violating the requirements for MaxEnt. Have you empirically tested the performance of IQL when the required assumption is not satisfied, i.e., demonstrations are according to (a)?

Correctness: Yes.

Clarity: Yes.

Relation to Prior Work: Yes.

Reproducibility: Yes

Additional Feedback:


Review 3

Summary and Contributions: update: rebuttal read. Overall, a good paper, but can be clarified in several places (see reviews). --- The authors reduce the IRL problem (with rewards + constraints) to a recursive linear problem over r(s, a) for rollout data collected from a Boltzmann distribution over the optimal Q-value function. Assuming terminal rewards are known, in a special case, the non-terminal rewards r(s, a) can be inferred. The authors formulate their approach in both the tabular and function approximation setting. They show mostly in the latter setting that their method get closer to the true r(s,a) / violates fewer constraints than baselines.

Strengths: The paper is written clearly and the diversity in experiments is nice.

Weaknesses: My main concern is how feasible this approach is in general. What if the data didn't come from the optimal (Boltzmann distribution) policy? What if there is a significant (suboptimal) bias in the policy? What if the optimal policy is multimodal (e.g., going left or right doesn't make a difference)? How robust is this method to discretizing continuous state-action spaces?

Correctness: Yes, as far as I can tel.

Clarity: Yes.

Relation to Prior Work: I'm not very familiar with all the recent IRL work, so I cannot judge this definitively.

Reproducibility: Yes

Additional Feedback:


Review 4

Summary and Contributions: Update: The rebuttal clearly addressed my concern on the restriction of Boltzman Distribution Assumption. However, R2 raised a new question [1]. I think the author needs more careful discussion about the proof of Theorem 1 in their rebutall, i.e. what if \pi(a) is zero for some actions, so that log(\pi(a)) is not defined. I strongly recommend the author reviewed their proof about this issue and add the discussion they had in the rebuttal to the main paper. I'd like to raise my rating to accept after reading the rebuttal. =================================================== Proposing an efficient IRL method that does not require costly inner-loop for estimating value functions, by exploiting an assumption that expert demonstration is following boltzman distribution of an underlying reward/value function. Experiments on simulators demonstrate its effectiveness compared to popular IRL baselines.

Strengths: An interesting solution backed up with math derivations. The authors also discuss for different situations/assumptions such as constrained IRL which I think provides insights to the community.

Weaknesses: I'm interested in the assumption that expert demonstration is generated under a boltzman distribution. Although this covers a broad range of problems, there're many other problem which may violate this assumption. For example, the expert may simply choose the max-reward action rather than sample actions from a distribution. I'm curious for the later case, do the authors think it's doable to adapt their method to this type of problems?

Correctness: Yes.

Clarity: I think in general the math part is a little bit tough to read. I hope the author can potentially make it more lightweight and provide more intuition / transition if possible.

Relation to Prior Work: Clear to me

Reproducibility: Yes

Additional Feedback:

[Author Response · NeurIPS 2020]

We thank all reviewers for their very constructive and valuable feedback and their insightful comments.

**Reviewer 1: The supplementary material does not contain the source code.** Due to confidentiality, we had to wait for permission to release code, which we got in the meantime. **We will release source code upon camera ready.**

**Reviewer 2, 3 and 4: One weakness of the approach is the underlying assumption that the expert demonstrations are generated by a Boltzmann distribution.** Most likely Eq. 1 in the paper led to some confusion here. To clarify: the expert demonstrations can follow an **arbitrary** distribution (including multimodal ones and greedy behavior choice), and need not necessarily follow a Boltzmann distribution. Our approach encodes this underlying arbitrary distribution of the expert in the long-term Q-value, such that a Boltzmann distribution over the estimated Q-function is equivalent to the original arbitrary expert distribution. **Put differently, our assumption is a restriction over the space of Q-functions, not expert policies** (see proof below). **We will clarify this and add the proof in the camera-ready.**

**Theorem 1.** *Define $Q^*(s,a)$ to satisfy the equation $Q^*(s,a) = Q^*(s,b) + \log(\pi^{\mathcal{E}}(a|s)) - \log(\pi^{\mathcal{E}}(b|s))$ for all actions $a,b \in \mathcal{A}$ (as in Section 3 ff.) and expert policy $\pi^{\mathcal{E}}(\cdot|s)$ of arbitrary underlying distribution. Then the Boltzmann distribution over $Q^*(s,\cdot)$ is equivalent to $\pi^{\mathcal{E}}(\cdot|s)$.* **Proof.** *The theorem follows from the inverse application of Equations (1)-(3). With $Q^*(s,a) = Q^*(s,b) + \log(\pi^{\mathcal{E}}(a|s)) - \log(\pi^{\mathcal{E}}(b|s))$ for all actions $a,b \in \mathcal{A}$, it follows from Eq. (3) that $\exp(Q^*(s,a)) = (\pi^{\mathcal{E}}(a|s)/\pi^{\mathcal{E}}(b|s))\exp(Q^*(s,b)) = \pi^{\mathcal{E}}(a|s)\sum_{A\in\mathcal{A}}\exp(Q^*(s,A))$ and thus $\pi^{\mathcal{E}}(a|s) = \exp(Q^*(s,a))/\sum_{A\in\mathcal{A}}\exp(Q^*(s,A))$.*

**Reviewer 2 and 4: What about a greedy expert policy?** We actually addressed the case of strictly optimal demonstrations in the SUMO experiments, Sect. 7.2. We further provide results for a greedy expert policy in Objectworld (Fig. 1 below). IAVI outperforms MaxEnt IRL also in this setting by multiple orders of magnitude w.r.t. runtime, and both IAVI and IQL yield a smaller EVD after less training time. **To be added to the camera-ready.**

| | IAVI | IQL | MaxEnt | SSMaxEnt |
|---|---|---|---|---|
| EVD | **0.22** | **0.21** | 2.8 | 2.96 |
| Runtime | **0.03 h** | **0.71 h** | 5.02 h | 4.65 h |

Figure 1: Results for a greedy expert policy in Objectworld. Visualization and table as in Figure 3 in the main paper.

**Reviewer 3: What if there is a significant (suboptimal) bias in the expert policy?** This is an open problem common to almost all published IRL methods which are based on the assumption of (soft-)optimal expert demonstrations, including e.g. GAIL or Maximum Entropy methods. We are aware that addressing systematic bias is important for future work, but out of scope for this work. Known bias could be accounted for by shifting the log probabilities of the original expert distribution. **We will add a comment to the camera-ready.**

**Reviewer 2: Are IAVI and IQL guaranteed to converge? What theoretical guarantees can you provide? Do they reach policy matching?** For expert policies with only non-zero action probabilities, we showed that the immediate reward function defined in Section 3 leads to a Boltzmann distribution over Q-values which reflects the expert policy. Since the Bellman update is a contraction, the reward values become more accurate with each iteration. Convergence guarantees for IAVI and IQL then follow from the convergence guarantees of Value Iteration and Q-learning under the same conditions. If there are actions with zero probability mass under the expert demonstrations, we add a very small conditioning term $\epsilon$ to the probability to avoid numerical instabilities. This leads us back to the case above with guaranteed convergence, but introduces a small deviation with respect to the match of the expert policy, bounded in dependence of $\epsilon$. Empirically, we showed convergence in our experiments in Section 7.1 und 7.2. **We will add more details to the camera-ready.**

**Reviewer 2 and 3: Can you extend IQL to Soft Q-learning or continuous action spaces?** Initial experiments with entropy regularization led to minor improvements. Our method can also readily be extended to continuous action-spaces. We currently evaluate a first working version. However, we regard these extensions as out of scope for the current submission. Results will be presented in future publications. **We will add a comment to the camera-ready.**

**Reviewer 2: Can't you calculate the action probabilities beforehand?** We formalized the more general case of an online algorithm, i.e. the possibility to add transitions during training. We explored the offline case of estimating the visitation probabilities beforehand in initial experiments with good results. **We will add this to the appendix.**

**Reviewer 4: The math is hard to follow.** **We will try to revise the math to add clarity in the camera ready**.

[Meta-Review · NeurIPS 2020]

Reviewers generally agreed that this paper proposes a novel IRL method that leverages the assumption that the expert demonstration is following a boltzmann distribution.